# Low Entropy Sub-Networks Prevent the Integration of Metabolomic and Transcriptomic Data

**DOI:** 10.3390/e22111238

**Published:** 2020-10-31

**Authors:** Krzysztof Gogolewski, Marcin Kostecki, Anna Gambin

**Affiliations:** Institute of Informatics, University of Warsaw, 02-097 Warsaw, Poland; mrckostecki@gmail.com

**Keywords:** genome-scale metabolic networks, information redundancy, metabolic landscapes analysis, graph entropy, renal cell carcinoma, transcriptomics

## Abstract

The constantly and rapidly increasing amount of the biological data gained from many different high-throughput experiments opens up new possibilities for data- and model-driven inference. Yet, alongside, emerges a problem of risks related to data integration techniques. The latter are not so widely taken account of. Especially, the approaches based on the flux balance analysis (FBA) are sensitive to the structure of a metabolic network for which the low-entropy clusters can prevent the inference from the activity of the metabolic reactions. In the following article, we set forth problems that may arise during the integration of metabolomic data with gene expression datasets. We analyze common pitfalls, provide their possible solutions, and exemplify them by a case study of the renal cell carcinoma (RCC). Using the proposed approach we provide a metabolic description of the known morphological RCC subtypes and suggest a possible existence of the poor-prognosis cluster of patients, which are commonly characterized by the low activity of the drug transporting enzymes crucial in the chemotherapy. This discovery suits and extends the already known poor-prognosis characteristics of RCC. Finally, the goal of this work is also to point out the problem that arises from the integration of high-throughput data with the inherently nonuniform, manually curated low-throughput data. In such cases, the over-represented information may potentially overshadow the non-trivial discoveries.

## 1. Introduction

Observing the technological progress of the recent decades one can notice that it facilitated an access to vast biomedical data resources describing different molecular levels (so-called -omics data). Consequently, our comprehension of biological processes becomes more profound as well as reliable. These facts open up a broad field of data integration, that aims to infer from various data collection platforms taking into account known biological dependencies between them.

**Motivations.** In the literature, it is broadly emphasized that the integration of *-omics* data improves understanding of various phenomena. The modern studies and corresponding literature emphasize the significant role of the integration of different *-omics* data types for better comprehension of a phenomenon of interest [1]. However, these integration procedures, thanks to a wider perspective, are not only meant to provide a new insight into a specific phenomena, but above all should constitute a deeper understanding of the general genotype-phenotype gap and the relationship between these two layers [2]. Importantly, each unique set of -omics data can be integrated using various statistical methods, and thus may result in an unprecedented outcome. As a consequence, each year a number of reviews are published to track and summarize the current state of the art in the field of -omics integration, which one can check for more detailed discussions [3,4].

In this work, however, the aim is to focus on a particular problem of the transcriptomic and metabolomic data integration. We report an interesting phenomenon related to the analysis of individual metabolic networks in the context of transcriptomic data. The conducted study allowed to detect a common and robust artificial pattern identifiable for each data cohort from the Cancer Genome Atlas that splits cancer-specific patients into two to four clusters. Interestingly, each cluster is characterized by a peculiar activity pattern among reactions hubs. In the article, we prove that these observations result mostly from the uneven metabolic information distribution within the metabolic network. To the best of our knowledge, no one has yet commented on the problem that the structure of a metabolic network can introduce when used along with integrative methods, in particular when used in the context of the flux balance analysis (FBA) [5].

**Related research.** One of the first attempts to this type of integration was suggested by Covert and Palsson, where authors infer the binary enzymatic activity from the transcriptomic continuous signal [6]. Next, a cascade of methods was proposed to harness transcriptomic data for analysis of metabolic networks. Among the most discussed methods we can point out: E-Flux [7], GIMME [8], GIMMEp [9], iMAT [10], INIT [11], MADE [12], mCADRE [13], PROM [14], RELATCH [15]. Apart from some minor details, fundamentally these approaches differ at three basic levels: (i) the way of inference of the enzymatic activity from transcriptomic data; (ii) the determination of the flow capacity boundaries from transcriptomic data; (iii) number of samples needed to create a mathematical model. These methods along with their other features, were already discussed and compared in few reviews where detailed descriptions can be found [16,17,18,19]. Therefore, here, we focus on a summary of what type of biomedical outcomes they have provided so far.

Using this type of models, where a general metabolic network becomes context-specific through integration with a transcriptome of specific tissue or organism few groups of researchers have already reported some interesting discoveries. With nearly 2000 samples of breast tumor Leoncikas et al. discovered a novel poor prognosis cluster characterized by local production of serotonin along with active extracellular matrix and membrane remodeling reactions [20]. Li et al. by integrating transcriptomic knowledge with human metabolic network suggest a supervised method to predict novel drug-target interaction [21]. In their work they predict related metabolic reactions and enzyme targets for approved cancer drugs, and predict drug targets with statistically high confidence rate. Reconstruction of a genome-scale metabolic models for 126 human tissues and cell types including healthy and tumor type was derived using the Recon 1 human metabolic network accompanied by transcriptomic data [13]. Among all, the set of models includes 26 tumour-specific models accompanied by their normal counterparts, in particular 30 models of brain tissue subtypes were determined. Next, using the modified version of iMAT, differential fatty acid uptake into mitochondria along with arachidonic acid and eicosanoid metabolism were suggested to explain different proliferative rates and invasiveness between PC-3/M (highly proliferative, cancer stem cells) and PC-3/S (highly invasive, epithelial-mesenchymal-transition-like properties) subpopulations derived from prostate cancer cell line [22].

In summary, the general biomedical aim of this approach is to model the gene–protein–reaction interactions in the form of a metabolic network and infer multiple biological properties, e.g., post-transcriptional gene activities or intensity and activity of metabolic reactions, that can explain the nature of analyzed sample.

**Our results.** In this paper, we want to draw attention to a specific problem related to the task of the data integration and analysis based on the general metabolic networks along with transcriptomic data. First, we present how the specific structure of sub-networks can inflate the importance of specific groups of enzymatic reactions, which may lead to incomplete or questionable conclusions. Next, in order to cope with that obstacle, we suggest a possible routine that can track and eliminate the unwanted problem influencing the metabolic knowledge obtained from the integration procedure. Furthermore, we present our results from the analysis of the TCGA renal cell carcinoma (RCC) dataset, which point out reactions discriminating patients in the context of the observed clinical factors. Additionally, we report the discovery of a poor-prognosis cluster of patients along with its characterization. Finally, we point out and discuss some already existing methods of analysis of metabolic networks that are likely to be influenced by this subtle yet significant network structural property. The preliminary version of this study was published as an extended abstract in the proceedings of the 2018 IEEE International Conference on Bioinformatics and Biomedicine (BIBM), pp. 96–101; Madrid, Spain.

## 2. Materials and Methods

### 2.1. The Human Genome-Scale Metabolic Reconstruction

In general, a metabolic network is a set of reactions among elements of a given set of metabolites. Each reaction may be associated with a specific genetic rule that needs to be met in order for reaction to occur. If a reaction has no genetic rule assigned to it, it can be triggered whenever all substrates are available. A genetic rule usually describes which genes/proteins need to be active/present in the system to carry out the reaction. In particular, a rule may require simultaneous presence of several enzymes (e.g., the transfer of L-Oh-Proline by the Apical Imino Amino Acid Transporters in Kidney and Intestine requires both Transmembrane Protein 27 and Solute Carrier Family 6), or alternative enzymes that can catalyze the reaction (e.g., efflux of 2-hydroxy-atorvastatin-lactone into bile that is supported by ATP binding cassette subfamily C member 2 or subfamily B member 1). Finally, each reaction has also a lower and upper bound describing minimal and maximal flow of metabolites through this reaction.

On the other hand, formally, the network can be considered as a Petri net with conditional transitions of places, where by a conditional transition we mean additional Boolean formula that needs to be met in order for transition to occur (see Figure 1 for the example).

In our study we used the human genome-scale metabolic reconstruction model, RECON 2.2 that details known metabolic reactions occurring in humans, and thereby holds substantial promise for studying complex diseases and phenotypes [23]. The model described 7785 reactions (i.e., transitions of a Petri Net), between 2654 metabolites (i.e., places), out of which 4742 depended on a genetic rule (i.e., condition of a transition). We will refer to these reactions as enzymatic reactions. Each genetic rule was a Boolean formula in a disjunctive normal form (DNF), where each Boolean variable corresponds to the activity state (active or not) of one of 1670 unique genes. Additionally, all of the metabolites in the network were distributed among 10 compartments. In total, we considered 6048 compartment-specific metabolites. For more details see Table 1 and Figure 2.

### 2.2. Metabolic Landscapes

In order to construct a sample-specific metabolic model from a general metabolic network given a transcriptomic pattern the following procedure was applied. First, we specified the thermodynamic conditions describing the acceptable capacity of reactions fluxes, i.e., set the minimal and the maximal flow levels through each reaction. Next, we evaluated each genetic rule, based on the transcriptomic data to decide which reactions could occur in the initial model (see an example in Figure 1). Finally, the obtained, individualized model could be further studied taking into consideration the metabolic network structure. In our work, the linear programming problem was formulated and the simplex algorithm was run to find the steady-state flux distribution. This procedure provided two types of information. First, it suggested a new pattern of fulfillment among the genetic rules (binary-valued), where each change, in contrast to the original value, could be considered as a post-transcriptional change. Second, it outputted a vector of fluxes that met the criteria of the linear problem that was solved. These fluxes could provide information about the metabolic reactions or pathways that were mostly exploited given such expression patterns.

### 2.3. TCGA Transcriptomic Data

In this study we used 20 RNA-seq transcriptomic datasets provided by TCGA, each composed of cancerous and control samples and accompanied by clinical data. For the purpose of our work, each dataset was subjected to a standard preprocessing routine with recount R package [24]. The raw counts were scaled by the total coverage of the sample, that is, the area under the curve (AUC) of the coverage. Next, only genes that were composing genetic rules in the RECON 2.2 model were selected. Finally a number of 5 read counts was selected to be a threshold for a gene to be considered as active. After Leucken et al. the thresholding procedures were as permissive as possible, so that downstream analysis could be conducted and interpreted [25].

In the presented case study we used the kidney cancer dataset (renal cell carcinoma, RCC) composed of the 897 primary tumor and 129 normal tissue samples. Tumor tissues were also classified by their morphological type subtype into four groups: 527 clear cell RCC, 290 papillary RCC, 66 chromophobe RCC and 14 unclassified RCC samples. For 201 patients the survival data were also available. Additionally, we used the brain cancer dataset (707 samples) to showcase the artifacts of the clustering results obtained from the metabolic landscapes.

### 2.4. Steady-State Flux Distribution

In order to determine the activity state of each reaction in a personalized model we made use of the approach proposed by Shlomi et al. [26]. We formulated a mixed-integer linear programming (MILP) problem (see Equation (1)) and solve it using Gurobi solver [27]. The goal was to maximize an objective function (1a) with respect to stoichiometric (1b), thermodynamic (1c) and transcriptomic (1d–g) constraints. For detailed description see the Methods section in [26].

The solution of the described problem provided a binary vector of activity states for each reaction which we refer to as a metabolic landscape.
(1)$$\begin{array}{ccc} \max_{v,y^{+},y^{-}}\sum_{r\in A}\left(y_{r}^{+}+y_{r}^{-}\right)+\sum_{r\in I}y_{r}^{+} & s.t. & \left(a\right)\\ S\cdot v=0 & \; & \left(b\right)\\ v_{min}\leq v\leq v_{max} & \; & \left(c\right)\\ v_{min,r}\left(1-y_{r}^{+}\right)\leq v_{r}-y_{r}^{+} & r\in A & \left(d\right)\\ v_{max,r}\left(1-y_{r}^{-}\right)\geq v_{r}+y_{r}^{-} & r\in A & \left(e\right)\\ v_{min,r}\left(1-y_{r}^{+}\right)\leq v_{r} & r\in I & \left(f\right)\\ v_{max,r}\left(1-y_{r}^{+}\right)\geq v_{r} & r\in I & \left(g\right)\\ v\in R^{m},\;\;y_{r}^{+},y_{r}^{-}\in \left\{0,1\right\} \end{array}$$

In the described setting the FBA problem was formulated as a linear program and solved using the simplex algorithm. However, it should be emphasized that the structure of the underlying graph was crucial in context of the flux analysis and the existence of dense sub-graphs loosely communicating with neighbouring vertices strongly influenced the final outcome of the algorithm.

### 2.5. Graph Entropy

To detect the possible sources of the structural redundancy in a metabolic network we applied the measure of entropy on a graph structure. To account for the context of the metabolic fluxes we made use of the entropy based on the inner and outer neighbours of vertices within a given sub-graph [28].

Namely, for a graph G=(V,E) and its sub-graph $G^{\prime }=\left(V^{\prime },E^{\prime }\right)$ where $V^{\prime }\subset V$ and $E^{\prime }\subset E$ contains all edges adjacent to vertices in $V^{\prime }$ we define the probability of the inner ($P_{I}$) and outer ($P_{O}$) flow of a vertex $v\in V^{\prime }$ as:$$P_{I}\left(v\right)=\frac{I(v)}{N(v)};\;P_{O}\left(v\right)=\frac{O(v)}{N(v)}=1-\frac{I(v)}{N(v)}$$
where I(v), O(v) are the number of neighbours of *v* that: belong to $V^{\prime }$ and not belong to $V^{\prime }$, respectively, while N(v) are all neighbours of *v*. Based on that we define the entropy H(v) of a vertex *v* as:$$H\left(v\right)=-P_{I}\left(v\right)\log_{2}P_{I}\left(v\right)-P_{O}\left(v\right)\log_{2}P_{O}\left(v\right)$$
and consequently, the entropy of a graph G induced by sub-graph G′, $H_{G^{\prime }}\left(G\right)$, is defined as:$$H_{G^{\prime }}\left(G\right)=\sum_{v\in V^{\prime }}H\left(v\right)$$

Given the above definitions, we determine the entropy of a group of enzymatic reactions within the RECON 2.2 metabolic network. Additionally, we evaluate the entropy induced by groups of random reactions sampled from the entire network. These steps aim to depict the graph in terms of the entropy and thus connectivity and the flow of metabolic information, that is subject to the metabolic landscape analysis.

### 2.6. Binary Data Analysis

Since the final outcome of a linear programming problem is a set of binary vectors, it is necessary to introduce specific methods to analyze them. In the literature there are described various statistical methods that are specifically dedicated to the analysis of binary data, including: the non-negative matrix factorization [29], the sparse logistic PCA method proposed by Lee et al. [30], and the variational factorization method employing independent beta latent densities [31].

Nonetheless, in our case, in order to explore the data, track the potential latent variables and visualize results we use the standard principal component analysis (PCA), which provides numerically stable and robust results and is also used to define a function for separating variables selection.

#### PCA Loadings-Based Variables Selection

Let $M\;\in \;{\left\{0,1\right\}}^{m\times n}$ be a binary data matrix with *m* observations and *n* variables, $L=(L^{\left(1\right)},\ldots ,L^{\left(n\right)})$ be a loadings matrix and *P* scores (or principal components) matrix, satisfying P=ML. By definition *P* is a representation of *M* in a new basis that is composed of vectors of *L*. For the purpose of binary landscapes analysis, we construct a set of highest valued coordinates from first *k* directions of the rotation matrix. The general parametrized function L describes the selection procedure:$$L_{f,k}\left(L\right)=\overset{k}{\underset{i=1}{⋃}}\left\{j\;:|L_{k}^{\left(i\right)}|\geq f\left(L^{\left(i\right)}\right)\right\}$$
where *f* is the function determining the threshold for the value of a coordinate to be assumed significant and *k* is the number of vectors from loading matrix that are considered. The function is used to select the groups of reactions with the highly correlated activity pattern between all samples. In parallel, the results from PCA Loadings-based variable selection were compared with the overall entropy analysis of the network structure considering the interaction between the groups of enzymatic reactions. It turned out that the groups requiring the adjustment belong to the group of clusters characterized by the low entropy and loosely connected to the other vertices in the network.

To determine differentiating variables in the studied datasets, two measures were used. First, the Jaccard Index is used to determine differentiating reactions between two groups of samples. For any $z\in {\left\{0,1\right\}}^{n}$ we define $z^{1}=\sum_{i}z_{i}$ and $z^{0}=\sum_{i}\left(1-z_{i}\right)$. Given that the Jaccard Index of two binary vectors $J:{\left\{0,1\right\}}^{n}\times {\left\{0,1\right\}}^{m}\to \left[0,1\right]$ is defined as:$$J\left(x,y\right)=\frac{\min\left(x^{1},y^{1}\right)+\min\left(x^{0},y^{0}\right)}{\max\left(x^{1},y^{1}\right)+\max\left(x^{0},y^{0}\right)}$$
The index is non-linear and depends on the lengths of vectors, thus evaluates to 0 when two vectors don’t share any element, increases with an increasing number of shared elements between vectors. It reaches the maximal value of 1 when both vectors share the same elements and the same lengths. In the process of variables selection as a threshold value we set 0.25 of the maximal value that the index can reach (given the lengths of compared vectors).

Additionally, the Tanimoto similarity measure of two binary vectors $T:{\left\{0,1\right\}}^{n}\times {\left\{0,1\right\}}^{n}\to \left[0,1\right]$ is defined as:$$T\left(x,y\right)=\frac{n-|\{i:\;x_{i}\neq y_{i}\}|}{n+|\{i:\;x_{i}\neq y_{i}\}|}$$
and is used as a distant measure between two binary metabolic landscapes.

## 3. Results

### 3.1. Metabolic Network Structure Problems

The first step was to apply the procedure suggested by Shlomi et al. to the cohort of the TCGA data in order to verify if there existed a common metabolic pattern among various types of tumors, or alternatively, if there existed any metabolic biomarkers which may be of diagnostic importance.

As mentioned in the introduction, our preliminary results suggested that, surprisingly, each data set was composed of two to four clearly separable clusters of patients. In the case of 13 out of 20 TCGA cancer datasets, the corresponding set of metabolic landscapes was well-separated into four clusters that, on average, were differentiated by the activity of 180 reactions. Additionally, for the remaining seven cancer-specific landscape datasets there were on average 95 reactions differentiating samples into two clusters (see Figure 3 for an example based on brain cancer dataset). Moreover, it turned out that almost all (>95%) discriminating (using $L_{\max,2}$ function) reactions in these clusters were coordinated by two main enzymes that were encoded by: *SLCO1A2* (solute carrier organic anion transporter family member 1A2; coordinating superfamily of 94 Amino Acid-Polyamine-Organocation reactions [32]) and *SLC7A9* (solute carrier family 7 member 9; coordinating superfamily of 79 Resistance-Nodulation-Cell Division reactions [33]) genes. These observations may lead to a conclusion that cancers in general have natural subfamilies that can be described by the activity of specific groups of metabolic reactions and thus also activity of particular enzyme encoding genes. Activation of *SLCO1A2* is related to the development and functioning of the immune system, organismal system for calibrated responses to potential internal or invasive threats and is highly over expressed in breast cancer tissues [34]. On the other hand, *SLC7A9* enables the transportation of substances (such as macromolecules, small molecules, ions) into, out of or within compartments of a cell, or between cells. Additionally, as a co-enzyme with *SLC3A1* coordinates group of five transportation/exchange reactions of L-Cystine, L-Alanine and L-Ornithine, that were also detected as differentiating the cancer data. This observation implies activation of *SLC3A1*, that was recently reported to promote breast cancer tumorigenesis [35].

Even though these literature reports may sound promising, we report another observation related to the analysis performed on a dataset with 500 randomly generated gene activities that were subjected to the metabolic analysis (see the right panel of the Figure 3). Despite the fact that the gene activity dataset did not include any relevant information, we were able to identify two groups of reactions differentiating samples into four well-separable clusters. These observations, undoubtedly, put in question all results related to the clustering of metabolic landscapes performed on all cancer datasets, since the analysis inferred about the knowledge that did not come from the data but from the topology and structure of metabolic network.

The analysis of the RECON 2.2 network structure revealed that there existed groups of reactions that were associated with the same genetic rule (top 10 most common genetic rules were related to over 900 reactions). Even though these reactions were biologically non-redundant (each of them was functional), they very often formed well-connected sub-networks that were loosely connected with the rest of the network. As a consequence they made it problematic to apply the FBA methods. As the way to identify such structures we proposed to apply the notion of entropy induced by sub-networks. It turned out that these sub-networks minimized the entropy.

In the Figure 4 one can notice the structural properties of the RECON 2.2 metabolic network. We discovered that among the groups of enzymatic reactions that follow the same genetic rule, there was a representation of these that consisted of several dozen of reactions and yet induced a very low entropy of the sub-network. The entropy- and structural-based analysis revealed that these reactions were very well connected within their group, however they were loosely connected to the neighbouring vertices outside their group. Such characteristics imposed the risk of the uneven flow of the metabolic information throughout the network and made it of low probability that initial transcriptomic conditions would be changed due to the influence of the other neighboring reaction states. In effect, this may result in the detection of network topology artifacts rather than new information on metabolic reactions.

For this reason we proposed two adjustment methods that take into account this phenomena. The aim was to transform the data so that the statistical analysis did not detect artificially induced data separation, but rather may result in a discovery of subtle differences in metabolic activity between samples possibly related to novel metabolic biomarkers.

### 3.2. Computational Workflow

Due to the discovery of the artifacts in the results of the metabolic landscape analysis we propose the following pipeline of procedures, that aims to reduce the statistical redundancy immersed in the RECON 2.2 network structure.

As the prerequisite, the entropy levels of each group of enzymatic reactions are determined with the tools described in the Materials and Methods section. These are the base of the control for redundant activity patterns in the statistical inference from the reaction activities.

First, we processed the RNA-seq transcriptomic data and converted it to the gene activity matrix using the threshold of five counts. The matrix was then used to create a personalized metabolic network model. For each of these, a MILP problem was formulated and solved with the Gurobi solver based on the simplex algorithm. The solution was composed of the sample-specific metabolic landscapes, which were subjected to statistical analysis. The analysis of the data was supported by the two-fold verification if the redundancy among groups of reactions existed. On one hand, using the PCA loadings-based variables selection L function defined in the subsection Binary Data Analysis, on the other assessing the entropy of the detected group of reactions. Based on these two conditions, if needed, the adjustment of these reactions was performed.

The data transformation was based on the aggregation of the activity states for groups of reactions with respect to compartments to which belong their substrates and products. Namely, each reaction was labeled with a name of form $s_{1}\ldots s_{k}$-$p_{1}\ldots p_{j}$, where $s_{1},\ldots s_{k}$ and $p_{1},\ldots ,p_{j}$ are alphabetically ordered names of compartments that, respectively, substrates and products belonged to. Next, all reactions with the same genetic rule and the assigned label were aggregated into the one represent reaction with the level of activity equal to the average of activities in the group.

Finally, inference from the transformed data structure through the hierarchical clustering using the binary distance measures (e.g., the Tanimoto similarity), correlation with the clinical data, selection of discriminatory features and functional analysis of determined clusters was performed. The outline of the described workflow is also depicted in the Figure 5. Additionally, in the Github repository: https://github.com/storaged/metabolic-landscape the code and the scripts used in this workflow are available. For the purpose of this study the workflow was executed on the machine with Mac OS.

### 3.3. Metabolic Landscape Adjustment for the Renal Cell Carcinoma

In order to validate the proposed workflow we performed the metabolic landscape analysis on the TCGA dataset of renal cell carcinoma (RCC). After performing the adjustment methods on the metabolic landscapes we reported a significant improvement in samples clustering, both in the sense of unwanted network structure-dependent clusters composition and correlation with clinical data (see Figure 6). Additionally, we took advantage of the provided prerequisite and confirmed that all of the adjusted groups of reactions fell into the group of the low-entropy level (see Figure 4).

We removed the amplified activity pattern of reactions coordinated by the same genetic rule, that influenced the clustering of samples in an unwanted way. After the reduction there were no significant, discriminating reactions associated with the same genetic rule. This step also resulted in more reliable clustering of data according to clinical observations, e.g., normal or tumor tissue type or morphological type of a tumor sample.

The results of our analysis of the renal cell carcinoma TCGA dataset were consistent with latest reports, that indicated *SLC6A3* as a experimentally confirmed biomarker for RCC [36]. The transcriptomic signal of *SLC6A3* in our data was clearly discriminating biomarker of Clear Cell RCC subtype (6.89 logFC).

However, thanks to the analysis of metabolic landscapes we further suggest potential biomarkers that correspond to specific, known RCC subtypes [37]. The literature so far reports *CXCL16* gene as a significantly expressed in papillary RCC with others still waiting for their validation [38]. Nonetheless, the analysis of metabolic landscapes suggests two transport reactions that discriminate the Papillary RCC subtype. Both of them are supported alternatively by the already reported *SLC6A3* or *SLC6A2*, other member of the same Solute Carrier family. The transportation activity state of dopamine and norepinephrine via sodium symport between cytoplasm and extracellular space are well separating the Papillary RCC subtype from other samples. Namely, these reactions are predicted to be inactive in Papillary samples. Our results reported 247 out of 291 (≈ 85%) papillary samples and 42 out of 737 (≈6%) other samples characterized by inactivation of these reactions. This observation was also confirmed by the purely transcriptomic data, which suggested down regulation of both genes (−3.62 and −2.2 logFC of *SLC6A2* and *SLC6A3*, respectively) for papillary samples. This observation suggested a simultaneous drop of the activity of both *SLC6A2* and *SLC6A3* as a potential diagnostic biomarker.

In case of the Chromophobe RCC (ChRCC) subtype, before the adjustment of the metabolic landscapes two genetic rules were in fact separating this subtype from the other cancer samples: (i) the extracellular space and cytoplasm exchange reactions supported by the complex of *SLC3A1* and *SLC7A9* (five reactions involving L- Cystine, L-Alanine and L-Leucine) and (ii) reactions controlled by *SLC7A9* that was involved in 79 reactions. However, after the adjustment two potential biomarkers were found: inactivation of sodium-dependent transport of (i) phosphate, supported by *SLC17A1* and (ii) ascorbate supported by *SLC23A1*. Inactive phosphate and ascorbate transport characterized, respectively, 65 and 62 out of 66 ChRCC samples; 45 and 28 out of 774 other tumor samples. Even in the literature reports it is still not clear how the phosphate transportation or concentration level and absorption via *SLC23A1* of ascorbate influences the cancer cells [39]. We point to these factors, both reaction activity and transcriptomic/proteomic levels, as possible biomarkers of ChRCC.

### 3.4. Poor Prognosis Cluster

Finally, we reported a candidate for a new cancer subtype resulting from the statistical and functional analysis of metabolic landscape clusters. Based on the samples clustering after the adjustment (see Figure 6) we further studied the obtained six clusters of samples. Among them, there were four homogeneous clusters composed mainly of: the healthy tissues (Control), the Chromophobe RTCC subtype (Chr-basal), the Clear Cell RTCC (CC-basal) and the Papillary RTCC (Pap-basal). Even though, the pattern of the remaining two clusters was not related to their morphological type we noticed a statistically significant (*p*-value: 0.002) difference in the survival time in one of the clusters (see Figure 7), which we addressed as a poor prognosis cluster.

The first and natural step here is to compare our clustering with the previously published [40]. In particular, the membership of the CpG island methylator phenotype (CIMP) cluster is of high interest, because of its bad prognosis properties that were investigated in [41]. It turns out that the entire CIMP cluster was part of the much larger (10 vs. 102) poor prognosis cluster that was identified based on its metabolic profile. Additionally, our poor prognosis cluster also contained five out of six samples classified as metabolically divergent (MD) in the literature. While CIMP-associated tumors showed increased expression of key genes involved in glycolysis, our cluster metabolic pattern was also associated with deregulations in glycan biosynthesis. However, the important characteristic feature was the reduced expression of genes related to amino acid transport.

To deepen the nature of these newly discovered clusters, we performed a differential analysis of four clusters (using also CC-basal and Pap-basal) in order to describe a metabolic as well as genetic nature of this cluster. We determined a set of 106 differentiating reactions coordinated by four genes: *OAT2* (3), *PRODH2* (4), *PKLR* (5) and *SLC1A2* (94), see Figure 4. Among these reactions we report orotate-glutamate antiport, uptake of allopurinol and oxypurinol by the hepatocytes, mitochondrial proline Oxidase (NAD) and dehydrogenase, and reactions involved in pyruvate metabolism. Genes coordinating these reactions reveal a specific expression pattern of the poor prognosis cluster, i.e., the low expression and the consequent inactivation of corresponding reactions. It is worth to emphasize, that the adjustment procedure provides a confirmation of the significance of the hub of reactions governed by the *SLC1A2* gene (see the corresponding datapoint in the Figure 4).

Additionally, we performed the functional analysis of top 100 differentiating genes, using the DAVID on-line tool. The analysis provided a consistent output indicating functions and keywords commonly related to modifications in transport and symport reactions highlighting transmembrane transport activity. Finally, pathway analysis performed with KEGG implied deregulations in the glycan biosynthesis and metabolism pathway.

The above observations can be preliminarily verified with literature reports. In a metabolic sense, abnormalities in glycan biosynthesis that we observe were reported as a significant factor of cancer cells phenotype and biology almost two decades ago [42]. It should be emphasized that recent studies stress the particular role of amino acid transporters in the pathogenesis of cancer. Specifically, the deregulation of these genes leads to metabolic reprogramming changing intracellular amino acid levels, which may underlie the molecular processes that explain poor prognosis in detected cluster [43]. Moreover, in our study, the poor prognosis properties can be influenced by the low activity of *SLC1A2* belonging to the organic-anion-transporting polypeptide (OATP) family, because it is responsible for transport of anticancer drugs (e.g., methotrexate used in chemotherapy) [44] and overall uptake of. Similarly, low expression of *OAT2* was reported to influence a poor response to antitumor UFT-based chemotherapy in colorectal cancer patients [45]. Loss of *PRODH2* was also reported in cancer [46], however no links with cancer prognostics were reported. The above summary may constitute an introductory justification for the possible existence of the poor prognosis cluster in RCC mainly conditioned by the chemoresistance dictated by the activity of its potential biomarker genes.

## 4. Discussion and Further Research

Concluding, in our article we recall an important yet barely discussed data analysis and integration problem of the two-fold nature. Using the TCGA datasets and the metabolic network model we presented how the bioinformatical and statistical data analysis may lead to outwardly interesting biological and medical observations, which may be justified by the literature reports. However, not only do we prove that these observations are the inevitable consequence of the model assumptions (as genetic rules induce clusters artificially), but also we show that these assumptions conceal the current state of knowledge about cell metabolism, as many discovery claims may testify.

We believe that the identified problem is important enough to look at alternative approaches and verify if they are not exposed to the similar obstacle. However, as it was already mentioned, there is a strong evidence that the problem highlighted in this work may be present in other researches that dealt with the analysis of the FBA outcome. The reason, even though is very subtle, is related to the linearity of the solving method applied to the non-linear topology of the problem support. In particular, the existence of internally strongly connected sub-graphs that are loosely connected with their neighbouring vertices influences the graph flow approaches. The metabolic information cannot be flawlessly passed on between reactions and thus interferenced by particular information-flow blockers. Here, we briefly discuss several alternative approaches as an extension to the literature overview presented in the Introduction.

To the scope of the traditional linear programming approach to solve the FBA problem we add what Lee et al. proposed in their work. The redefinition of the main objective function as well as an emphasis put on the more informative nature of absolute gene expression measurements provided by RNA-seq data was presented. Authors set an optimization problem to maximize the correlation between an observed transcriptomic profile and levels of reactions fluxes, that is solved using the MILP solver. Authors provide a case study based on estimation of exometabolic flux in *Saccharomyces cerevisiae* and show that their method outperforms the traditional approach upon maximisation of the rate of biomass production.

Additionally, there is also a bunch of approaches that extends the approach to the FBA problems onto the field of Bayesian statistics. One example of an extension of the FBA via Bayesian factor modeling was suggested by Angione et al. [47]. The extension bases on an assumption that high-dimensional data are generated from the hidden lower-dimensional factors that are shared across data samples. As a consequence, as a first step authors solve the bi-level FBA problem, which extends the standard problem by interdependencies among enzymes and genetic rules coordinating metabolic reactions. Next, Bayesian matrix factorization modeling with Gaussian Markov random field is incorporated to perform pathway analysis that takes into account reaction-pathway memberships as prior knowledge. Using an metabolic model of *Escherichia coli* authors present how their model tracks changes in pathway responsiveness for variuos experimental conditions.

Lately, in [48] authors presented an approach that results in flux posterior represented as an unimodal truncated multivariate normal (TMVN) distribution. Using a MCMC Gibbs sampler implemented in Matlab a group of in-silico experiments was performed to highlight the capabilities of the model. The presented approach allows to characterize the genome-scale flux covariances, reveal flux couplings, and determine genome-scale number of intracellular unobserved fluxes in *Clostridium acetobutylicum* from 13C data based on a small set of intracellular flux measurements.

Interestingly, the problem of flux estimation was also approached from the perspective of the thermodynamic analysis. Zhu et al. introduce a novel, two-step optimization method termed as thermodynamic optimum searching [49]. First, the original FBA is used to determine the maximum growth rate. Then, the most thermodynamically favored solution is acquired by solving a nonlinear optimization problem in which the growth rate is fixed to the maximum. The later aim is achieved by maximizing the entropy production rate while minimizing energy usage and deviation from the second law of thermodynamics related to the the minimum magnitude of the Gibbs free energy change and the maximum entropy production principle. The method is supported with five *E. Coli* case studies presenting the improved accuracy of predictions compared to the standard FBA.

What should be emphasized here is that none of the recalled articles mentioned any observation on the relationship between the structure of the metabolic network and their final outcome. Additionally, an alarming fact is the use of methods such as the categorization of the transcriptomic signal, linear (or semi-linear) methods of solving the flow problem, binarization of data and their interpretation based by clustering. All of these observations allow the reader to ask if any control for artifacts was performed. In order to compare the methods, approaches, objectives of all the mentioned works in this article a table summary is provided in the Appendix A.

To conclude, the future directions in which we would like to conduct this research include incorporating other metabolic networks for the landscape analysis, but also improved adjustment methods, that would define and take into account the confidence of the reaction. One idea in that direction, would be to incorporate some experimental validations from the literature reports. Ideally, we would like to reach the gold standard using measurements from the 13C-fluxomics technology, where metabolic precursors enriched with 13C and quantified by mass spectrometry or NMR [50,51], as well as from transcriptomic assays. However, such data has so far been collected for relatively small systems, e.g., single pathways [52] or other simple organisms [53], while our amendment concerns the whole human cell metabolism. Another option is to take into account the entropy-based approach and propose a sample-specific adjustment method. Finally, we also suggest defining a probability of reaction activity, that may be introduced instead of currently considered binary activity state. This would allow to make use of the continuous information included in the expression data, rather than reduced binary signal. We believe that these improvements may lead to better understanding of cancer biology and phenotype resulting from the observed differences in metabolic activity.

## Figures and Tables

**Figure 1 entropy-22-01238-f001:**
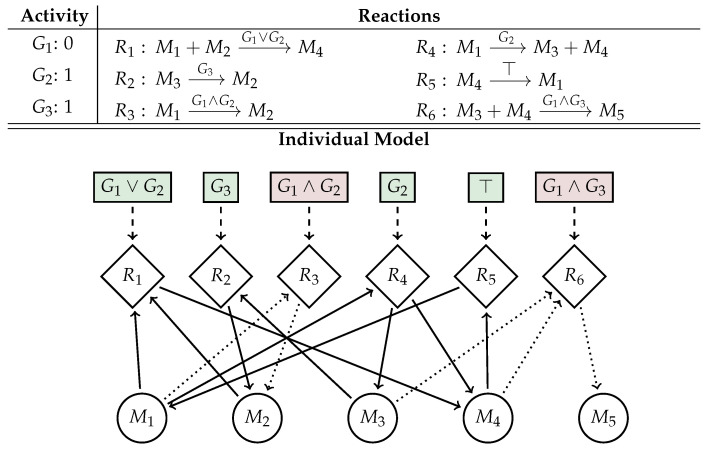
An example of metabolic network with genetic rules. The example is composed of six reactions $R_{i}$ among five metabolites $M_{j}$. Among these reactions there are five enzymatic reactions that are coordinated by the enzymes related to the three specific genes $G_{k}$. All arrows describe the flow of metabolites through reactions according to the above list of reactions. Using the gene activity pattern, a general network can be turned into transcriptom specific and represented as a Petri net with conditional transactions. Here we can see that inactivity of gene $G_{1}$ results in silencing (dotted line) of the reactions $R_{3}$ and $R_{6}$ which represses the production of the metabolite $M_{2}$ and eliminates the production of the $M_{5}$ in the system.

**Figure 2 entropy-22-01238-f002:**
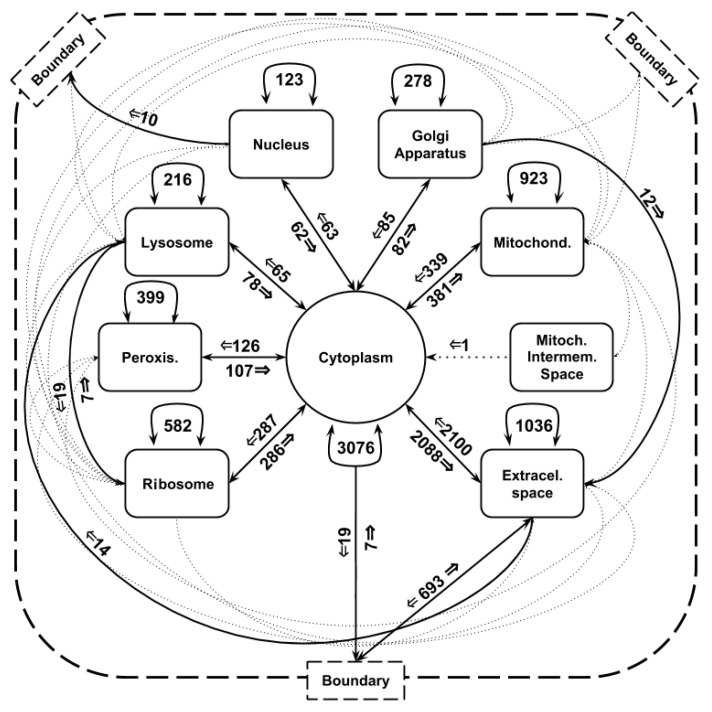
The figure presents an outline of the reactions distribution in the metabolic model of RECON 2.2. There are nine inter-cellular compartments depicted and the additional boundary component which represents external entry and exit of metabolites into the metabolism network model. Each line connecting two compartments represents all reactions that involve metabolites from these compartments. The number of these reactions and their direction is assigned to each line. To keep the figure readable, for less than 10 reactions we use a dotted, thinner line.

**Figure 3 entropy-22-01238-f003:**
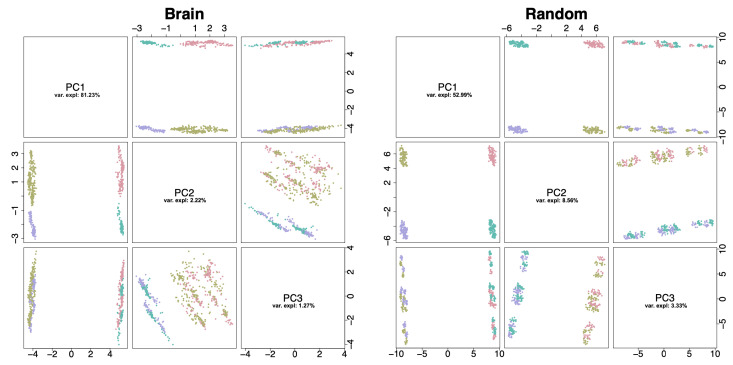
The comparison of the first three principal components of metabolic landscapes determined for brain cancer (**left**) and random (**right**) datasets. In both cases samples form well-separating clusters that can be identified by the activity pattern of overrepresented gene rules. For the brain dataset: *SLC7A9* and *SLC28A3*. For the random dataset: *SLC7A6* and *SLCO1B1*.

**Figure 4 entropy-22-01238-f004:**
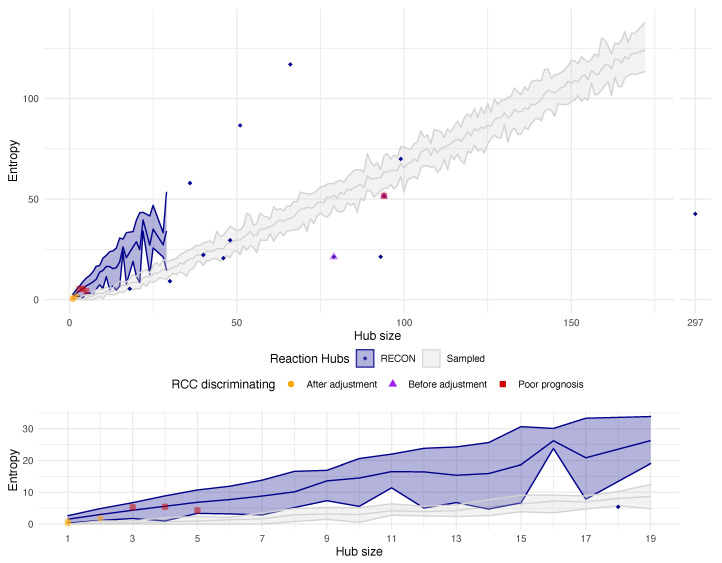
**The entropy levels induced by specific sub-networks in RECON 2.2 by their size**. The upper panel depicts the entropy level induced by each sub-network of enzymatic reactions following the same genetic rule (the blue area and outlying blue points). The grey area shows the entropy levels for random groups of reactions of a given size. The groups of reactions that characterize the RCC samples before and after the adjustment, as well as, the potential biomarker reactions for the discovered poor-prognosis cluster are depicted by corresponding points. The bottom panel is a close-up view of the low-number groups of reactions.

**Figure 5 entropy-22-01238-f005:**
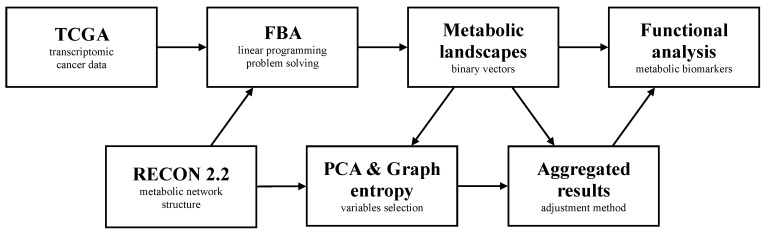
The workflow outline highlighting the main steps in the proposed data analysis method.

**Figure 6 entropy-22-01238-f006:**
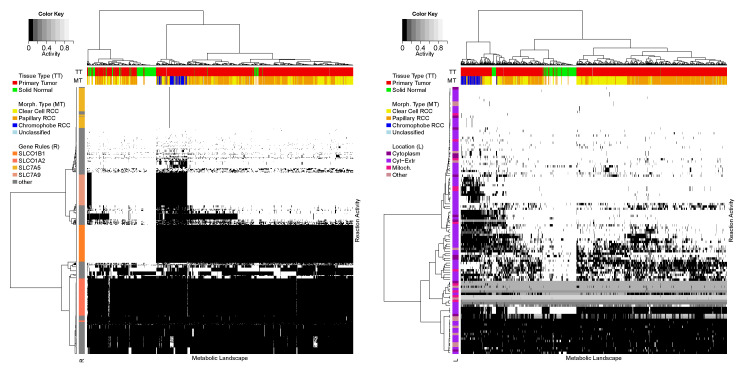
The comparison of reactions activity before (**left**) and after (**right**) the adjustment. On the left panel, the vertical strip marks reactions associated with the same genetic rule (the orange scale colors) noticeably determining the clustering of all landscapes. In both panels, horizontal stripes represent the Tissue Type (TT) and the Morphological Type (MT) of all samples. One can see, how the adjustment improves the correlation of the data with the clinical variables, especially the morphological type. Finally, the vertical strip on the right panel presents that correspondence related to the compartments (the purple scale colors) was introduced.

**Figure 7 entropy-22-01238-f007:**
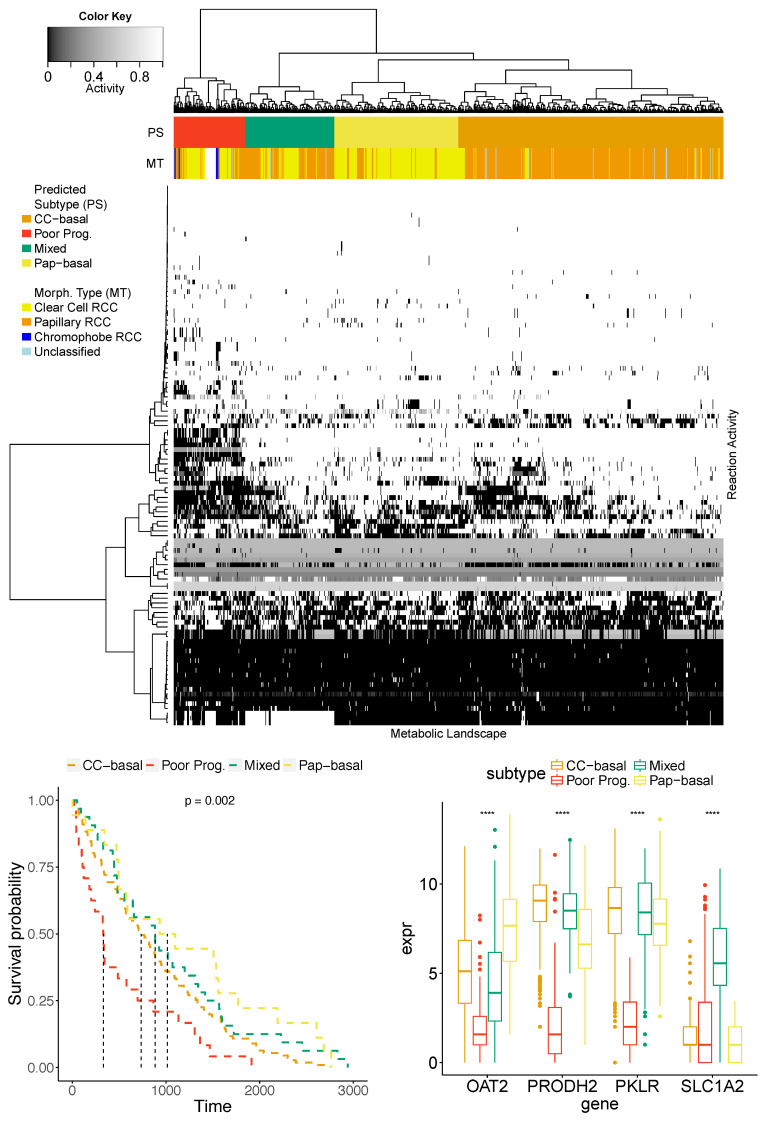
**The analysis of the poor prognosis cluster.** The heatmap presents the activity of reactions from the four clusters determined with the hierarchical clustering. The upper horizontal stripes compare the predicted subtypes (PS) labeling with the known morphological types (MT). The bottom-left panel presents the Kaplan–Meier survival curves, which present the significantly lower survival time of patients from the poor prognosis cancer (*p*-value: 0.002). The bottom-right panel compares the expression level of the genes characterizing the poor prognosis cluster.

**Table 1 entropy-22-01238-t001:** The table provides a summary of the RECON 2.2 metabolic model.

all reactions	7785	enzymatic	4742
		transport	3043
		exchange ${}^{a}$	701
		demand ${}^{b}$	44
		reverse	3782
		directed	4003
all metabolites ${}^{c}$	6048	unique substances	2654
		boundary	722
		compartments	10
all genetic rules	4742	simple rules	2912
		complex rules	1830
		unique rules	1341
		unique genes	1670

${}^{a}$ Exchange reactions describe in- and outflow of metabolites through the system boundary. ${}^{b}$ Demand reactions are intra-network, unlimited sinks or sources of metabolites degradation or production. ${}^{c}$ An additional metabolite (in contrast to the summary reported in [23]) was required for the *R_biomass_other* reaction.

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
