# Peer review of "Low Entropy Sub-Networks Prevent the Integration of Metabolomic and Transcriptomic Data"

_entropy, 2020, doi:10.3390/e22111238_

Round 1
Reviewer 1 Report
The overall premise of the article resonates well with my understanding of data integration challenges. The analysis and associated discussion is robust and well formulated. I enjoyed the frank discussion of how artefacts may arise in seemingly robust computational methods, as well as how to perceive the deficiencies and correct them. I have no qualms about accepting the paper for publication as it may be of substantial value in rigorous biomarker inference.
I am going to make one semi-technical comment, though, and say that I found it odd (from my background) to see a manuscript of this nature that somehow does not use either the word "error" or the word "bias" even a single time. Obviously there are indications of how control data are integrated into the analysis, which presumably enables the study to objectively handle some degree of random and directional imprecision (i.e.,, 'error' and 'bias', respectively), and I suspect that a careful reading of the methods and discussion sections could shed light on roughly how robust the modified schemes are in dealing with varying levels of data imperfection. However, for the sake of readers doing a quick skim (or, in order to facilitate the inevitable key word searches), it might be helpful for the authors to comment on model sensitivity and robustness.
Beyond this, I noticed a few minor typographical or grammatical errors:
discuss on some already
>> discuss some already
Fig. 1: 'Rybosome' should be Ribosome.
There were also used two measures to determine differentiating variables in the studied datasets...
>> To determine differentiating variables in the studied datasets, two measures were used...
they very often form a well-connected sub-networks that
>> they very often form well-connected sub-networks that
problem support.n particular,
>> problem support. In particular,
Reviewer 2 Report
Summary:
The study by Gogolewski, Kostecki, and Gambin describes common pitfalls relating to the structure of the metabolic network during the integration of metabolic data with gene expression datasets. They found that the sub-networks specific structure could inflate the importance of specific enzymatic reactions, leading to over-weighted features during clustering. The problem can be identified by examining the graph entropy distribution of the RECON 2.2. They describe a workflow which transforms the data so that the statistical analysis does not detect the artifact. While the idea is interesting, there are significant problems with the writing and organization of the manuscript. I also have doubts about the significance of their identified poor prognosis cluster.
Problem with the result. The hierarchical clustering result from figure 5 was not well described. I believe there might be more normal samples being clustered together after adjustment, but the author should quantify what was used to determine the correctness for these clusters. In normal gene expression analysis, differentiating normal vs. tumor are generally considered trivial, so I have a hard time seeing the significance of this result.
In figure 6, the author predicts the subtype of cancer. Here, the author should consider comparing their defined subtypes with previously published data like [PMID: 29617669] and [PMID: 26536169]. For example, from the literature, CpG Island Methylator Phenotype CIMP subtype is likely a better definition of renal cell prognosis. How does the known CIMP subtype compare to the author's poor prognosis group?
I applaud the author's effort to make their software available on Github. However, when I review the code, the input and output for each program were not well documented. The author should consider repackaging their program into a usable workflow with clear instructions defining your software's input and output.
Problem with the writing. The abstract and introduction need to be explicit with its definition of the problem. Line 237 describes the necessity for two-fold verification if the redundancy of reactions exists; however, it is not clear based on the writing how the redundancy check was performed. I think adding a figure for the workflow, in the beginning, would more smoothly guide the reader through their problem. I also have a problem with their statement that they are integrating metabolomics with transcriptomic. In reality, this is a gene expression analysis that attempts to infer reaction/enzymatic activity. I believe the word inference should be used in place of integration.
Reviewer 3 Report
The authors present a new approach for integration of multi-omics data sets to address the challenges of FBA-based methods. The authors identify a mathematical correction approach to improve metabolic landscape analysis. Overall, the manuscript describes sufficient background information and presents some promising results, but there are some areas of suggested improvement. Revisions and comments are suggested below:
Suggested revisions:
- Line 102: The authors should add a sentence briefly describing RECON 2.2 and how it is used in the field?
- Line 127: Could the authors explain the use of the 5 read count threshold? A citation or short explanation should suffice.
- Figure 3: For the PCA loading plots, the authors should include the percentage of variance on the axes, especially considering these were used for variable selection.
- Lines 181-196: The authors describe the influence of several genes from the brain data set, but it is unclear how the authors specifically identified how these are responsible for separating the 2-4 clusters of patients. Could the authors provide a sentence or two to describe what thresholds were used for the Jaccard Index and Tanimoto similarity measure to pinpoint differentiating genes?
- Figure 4: The hub size is limited for the RECON reaction hubs, but there are corresponding blue data points for this data set at up to hub size 100. Why does the blue area not extend further? Also, the poor prognosis cluster is well-grouped besides one outlier at hub size 90. Could the authors address why this reaction was not minimized after correction while the others were?
- Line 254: The authors “report a significant improvement in samples clustering” but do not provide a threshold for significance. The improvement is apparent in Figure 5, but the authors should specifically address what threshold was used. Can you quantify how the correlations are improved between the left and right panels?
- Line 258: The authors “remove the amplified activity pattern of reactions coordinated by the same genetic rule”. Please expand and clarify how the selection of these occurred? Specifically, was it the same subnetwork or literally the same Boolean rule?
Grammatical revisions:
- It may be beneficial to review the entire manuscript for word choice and grammatical errors.
- Line 22: “data taking into account” does not make grammatical sense; we suggest revision to “data collection platforms” or similar
- Lines 98-99 and Line 103: Use of parenthesis “()” should be consistent for clarification.
- For example, line 99 says “transitions (reactions),” but the line 103 says “reactions (transitions)” Same for “gene rules or genetic rules/conditions.”
- Line 102: It may be beneficial to reference Table 1 at the end of this sentence
- Line 386: “base” should be changed to “estimate based”
- Line 406: reword to “A literature search of articles was conducted, and Table A1 summarizes…”
- Line 408: reword to “Yet, artifact detection is not considered in any of the articles”
Round 2
Reviewer 2 Report
Thanks for the response.
Regarding the author's statement of "Our task is to compare metabolic landscapes, not gene expression which correlates with known histological subtypes." I apologize if I misunderstood the author's message in my original review. If that's the case, ideally, the study should utilize a gold standard dataset with inferred metabolic activities such as mass spec measurements of the metabolites. Ideally, there should be a more convincing validation of the author's model. I would recommend including one experimentally supported example to validate that their adjusted model is correct.
Figure 6 shows that the clustering of normal vs. tumor theoretically should be trivial for both metabolomic and expression data. Given that their pipeline was derived from binarization of the transcriptomic information, I still have concerns that the normal samples did not cluster together. My interpretation is that the clustering is likely due to noise or the elimination of the biological signatures introduced during the data processing. I am not convinced that some control samples are resembling tumors for some metabolic features.
Thank you for the update on the GitHub. While testing the program's software, the program failed to run, the expected input for the shell script is not correct. I believe more work needs to be done on this to make your program useful to the community. Also, based on the input file, only the binarized input is included. It will be great if the author could start with the raw quantified gene expression matrix and start with the binary transformation process. I also felt that more details about the output of the program are necessary.
